biochemistry/molecular biology

capping, non-canonical capping, RNA processing

**Author for correspondence:**
Yulia Yuzenkova
e-mail: yulia.yuzenkova@ncl.ac.uk

One contribution to the Life Sciences New Talent special collection.

# The expanding field of non-canonical RNA capping: new enzymes and mechanisms

Jana Wiedermannová[1], Christina Julius[2] and Yulia Yuzenkova[1]

[1]Medical School, NUBI, Newcastle University, Newcastle upon Tyne, UK
[2]Umeå University, Umeå, Sweden

JW, 0000-0002-9979-0799; YY, 0000-0003-4036-9235

Recent years witnessed the discovery of ubiquitous and diverse 5′-end RNA cap-like modifications in prokaryotes as well as in eukaryotes. These non-canonical caps include metabolic cofactors, such as NAD$^+$/NADH, FAD, cell wall precursors UDP-GlcNAc, alarmones, e.g. dinucleotides polyphosphates, ADP-ribose and potentially other nucleoside derivatives. They are installed at the 5′ position of RNA via template-dependent incorporation of nucleotide analogues as an initiation substrate by RNA polymerases. However, the discovery of NAD-capped processed RNAs in human cells suggests the existence of alternative post-transcriptional NC capping pathways. In this review, we compiled growing evidence for a number of these other mechanisms which produce various non-canonically capped RNAs and a growing repertoire of capping small molecules. Enzymes shown to be involved are ADP-ribose polymerases, glycohydrolases and tRNA synthetases, and may potentially include RNA 3′-phosphate cyclases, tRNA guanylyl transferases, RNA ligases and ribozymes. An emerging rich variety of capping molecules and enzymes suggests an unrecognized level of complexity of RNA metabolism.

## 1. Introduction

Cell fitness is highly dependent on fast and adequate changes of gene expression to cope with changing conditions. Gene expression is mainly regulated by adjusting RNA levels by production or degradation, but also by the differential utilization of RNA as defined by its sequence, structure or epitranscriptomic modifications. RNA capping is an important mechanism, affecting the overall fate of the respective RNA. Canonical eukaryotic 5′ RNA caps are attached to the nascent RNA early during its synthesis by RNA polymerase (RNAP) and can subsequently affect almost all its cellular roles. They

**Table 1.** List of abbreviations used in the text.

| list of abbreviations | | | |
| --- | --- | --- | --- |
| ADP | adenosine diphosphate | NC | non-canonical |
| ADPR | adenosine diphosphate ribose | NGD | nicotinamide guanine dinucleotide |
| ATP | adenosine triphosphate | NMN | nicotinamide mononucleotide |
| CoA | coenzyme A | NMNAT | nicotinamide mononucleotide adenylyltransferase |
| FAD | flavin adenine dinucleotide | NPnNs | dinucleotide polyphosphates |
| GDP | guanosine diphosphatase | PARP | poly (ADP-ribose) polymerase |
| GMP | guanosine monophosphatase | ppp-RNA | triphosphorylated ribonucleic acid |
| GTP | guanosine triphosphatase | pp-RNA | diphosphorylated ribonucleic acid |
| GTase | RNA guanylyltransferase | p-RNA | monophosphorylated ribonucleic acid |
| GTPase | guanosine $5'$-triphosphatase | PRNTase | polyribonucleotidyltransferase |
| $m^7G$ | $N^7$-methyl guanosine | RdRp | RNA-dependent RNA polymerase |
| MTase | methyltransferase | RNAP | RNA polymerase |
| NAAD | nicotinic acid adenine dinucleotide | TPase | RNA triphosphatase |
| $NAD^+$ | nicotinamide adenine dinucleotide | UDP-Glc | uridine diphosphate glucose |
| NADH | reduced nicotinamide adenine dinucleotide | UDP-GlcNAc | uridine diphosphate N-acetylglucosamine |
| NAM | nicotinamide | VPg | viral protein genome-linked (VPg) |
| NaMN | nicotinic acid mononucleotide | VSV | vesicular stomatitis virus |

enable initiation of protein synthesis, serve as an identifier for recruiting protein factors for pre-mRNA splicing, polyadenylation and nuclear export, and affect stability and susceptibility of RNA to nucleases [1].

Until recently, it was assumed that just eukaryotic cells are endowed with a $5'$ RNA caps, a $5'$-$5'$ linked $N^7$-methyl guanosine $(m^7G)^1$ added co-transcriptionally to the $5'$ ends of transcripts producing $m^7G$-RNA [2,3]. $5'$ ends of prokaryotic RNAs were considered less heterogeneous. $5'$ triphosphorylated RNA (ppp-RNA) results from the incorporation of nucleoside triphosphate as the initiating substance of the nascent RNA by RNAP. Prokaryotic ppp-RNA was supposed to be diversified only by removing some of its phosphates (becoming diphosphorylated, monophosphorylated or hydroxylated). The phosphorylation state of RNA $5'$ end defines RNA longevity and susceptibility to degradation, therefore affecting gene regulation and cell survival. Monophosphorylated transcripts are preferred substrates of the main degradation endonucleases (e.g. RNase E in *Escherichia coli* or the $5'$ exonuclease RNase J in *Bacillus subtilis*), which makes them vulnerable to rapid attack by either of them [4,5].

Recent years revealed several surprising facts about bacterial $5'$ RNA ends, some of them with consequential findings in eukaryotes. Besides the high percentage of diphosphorylated RNA (pp-RNA; 35%–50% mRNA) [6], a surprising number of different cap-like structures, including dinucleotide analogues, metabolic cofactors and cell wall precursors, were revealed at the $5'$ ends of bacterial and eukaryotic RNAs [7–10]. The discovery of bacterial non-canonical (NC) caps challenged the perception of eukaryotic RNA $5'$ end uniqueness [8,9,11–14] blasting off the new research field of RNA $5'$ end epitranscriptomics.

## 2. Canonical $5'$ RNA caps and capping mechanism

All known eukaryotes synthesize $m^7G$-RNA by the enzymatic activities of three enzymes: RNA triphosphatase (TPase), RNA guanylyltransferase (GTase) and guanine-$N^7$ methyltransferase (guanine-

---

[1]Abbreviations used in the text are summarized in table 1. Known and potential canonical and NC $5'$ RNA capping enzymes and their NC-caps are summarized in table 2.

**Table 2.** Summary of the known and potential canonical and non-canonical 5′ RNA capping enzymes described in the text and their 5′ RNA cap products (for details see the text).

| non-canonical 5′ RNA capping enzyme | cap |
| --- | --- |
| multi-subunit and single-subunit RNA polymerase, primase | variety of dinucleotides and nucleotide analogues (e.g. $NAD^+$, NADH, FAD, UDP-Glc, UDP-GlcNAc, $NP_nN$, CoA) |
| GTase | $m^7G$ |
| RdRp L protein of VSV | $m^7G$ |
| RNA ligase | ADP |
| tRNA synthetase (LysU) | $Ap_4N$ |
| RNA phosphate cyclase (RtcA) | ADP |
| PARPs | ADPR |
| NMNATs | $NAD^+$; capping of RNA not proved |
| CD38 | ADPR |
| ribozymes | $NAD^+$, FAD, CoA |

$N^7$ MTase) [1]. RNA TPase removes the γ-phosphate from the 5′ triphosphate to generate 5′ diphosphate RNA (figure 1a(i)). GTase transfers a GMP group from GTP to the 5′ diphosphate via a lysine-GMP covalent intermediate (figures 1a(ii), (iii) and 2a). The guanine-$N^7$ MTase then adds a methyl group to the $N^7$ amine of the guanine cap to form the cap 0 structure. +1 nucleotide of the resulting $m^7G$-RNA can be further methylated at the 2′-O-ribose of the first (cap 1) or first and second nucleotides (cap 2) [1]. Hypermethylation of 5′ $m^7G$ cap (cap 4 structure) can be found in trypanosomes and other Kinetoplastida [15]. A different methylation level further diversifies the function and localization of capped RNAs [1].

The described canonical capping mechanism is common for eukaryotic organisms as well as some DNA viruses (e.g. vaccinia virus) and double-stranded RNA viruses (e.g. reoviruses). By contrast, vesicular stomatitis virus (VSV), and probably other non-segmented negative-strand RNA viruses (Mononegavirales), use a different approach to cap RNA [16]. Its multifunctional RNA-dependent RNA polymerase (RdRp) L protein carries out an unconventional mechanism that involves the stepwise action of guanosine 5′-triphosphatase (GTPase) (figure 1b(i)), followed by RNA: GDP polyribonucleotidyl transferase (PRNTase) (figure 1b(ii), (iii)). The main difference is RdRp L protein-mediated transfer of the 5′-monophosphorylated pre-mRNA (figure 1b(ii)) to GDP through a covalent enzyme-phosphorylated RNA intermediate (figures 1b(ii,iii) and 2f). Interestingly the transfer mechanism involves the formation of a phosphoramide bond to histidine instead of lysine as occurs in the conventional system (compare figure 2a versus f). The sequence of methylation events in conventional versus VSV systems has a reversed order [16].

# 3. A great variety of non-canonical 5′ RNA caps was discovered in bacteria, eukaryotes and their mitochondria

First indications of a vast spectrum of 5′ RNA modifications were published in two mass spectroscopy-based papers from 2009, which screened for small-molecule conjugates of RNA in E. coli and Streptomyces venezuelae [8,9]. Besides NAD- and CoA-linked RNA, they identified succinyl-, acetyl- and methylmalonyl-thioester derivatives of CoA. All these metabolites were proved to localize at the 5′ end of cellular RNAs and can be therefore classified as 5′ NC caps. Authors of these studies also identified 3′-aminoacyl adenosine monophosphates (originated most likely from aminoacylated tRNAs, but their existence on other RNA species is not excluded) and 17 [9] or 24 [8] additional unknown species attached to unspecified sites of RNA, suggesting an unprecedented range of bacterial RNA modifications in vivo. The existence of NAD-RNA was confirmed in bacteria, yeast, human and plant cells, as well as in mitochondria [11,12,17–20]. Other NC caps were later identified in vivo—flavin adenine dinucleotide (FAD), uridine diphosphate glucose (UDP-Glc), uridine diphosphate N-acetylglucosamine (UDP-GlcNAc) and dinucleotide polyphosphates ($NP_nNs$) [7,10,14,21].

(a)

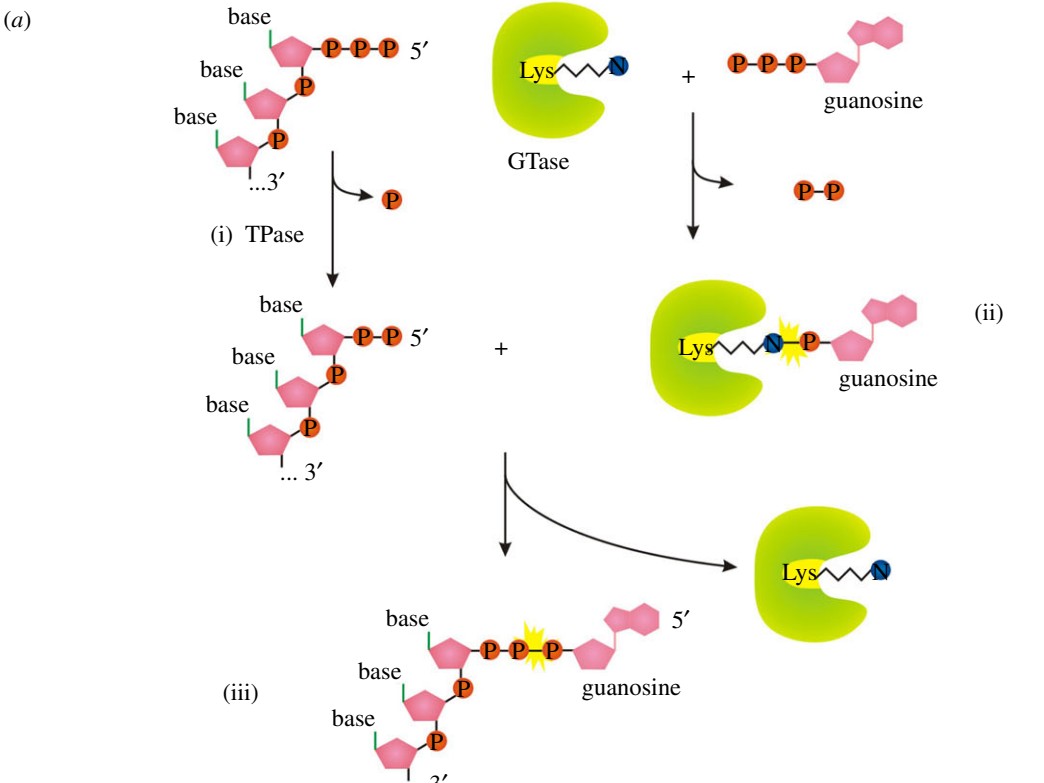

(b)

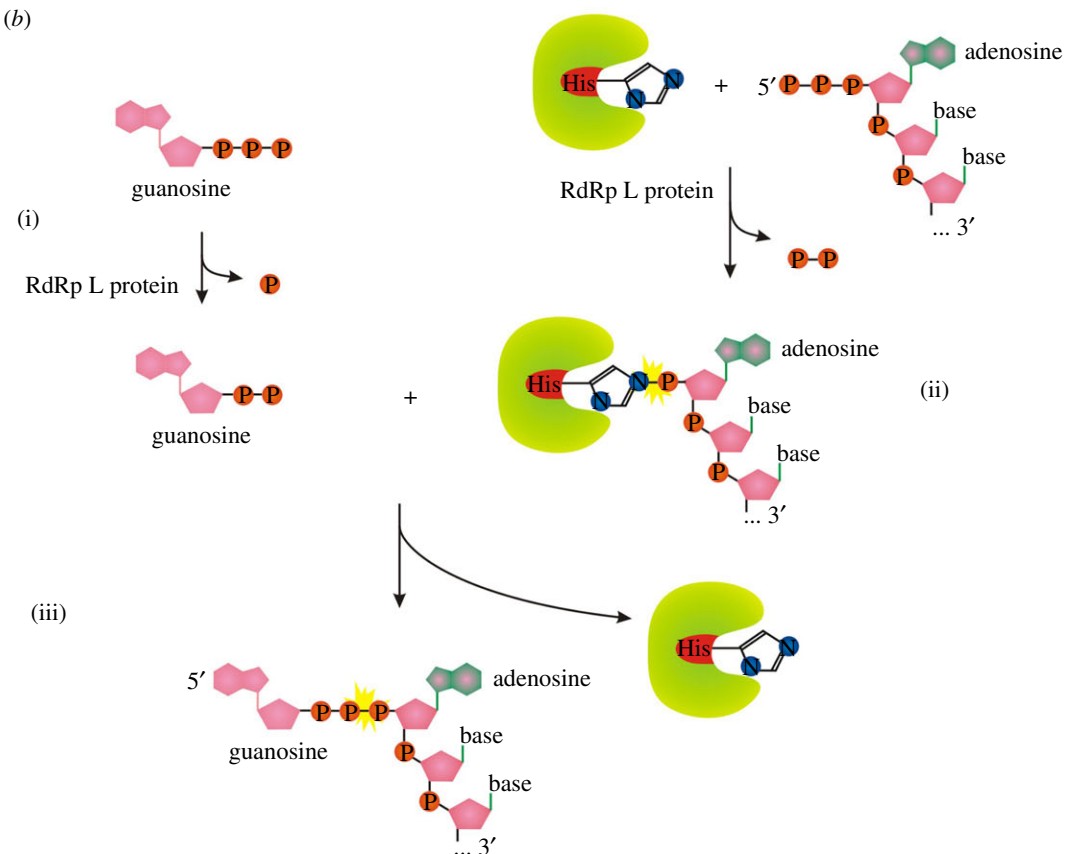

**Figure 1.** (a) Canonical mechanisms of m$^7$G cap formation in eukaryotes. (b) Alternative pathway in vesicular stomatitis virus. Orange 'P' symbolizes the phosphorus group, adenosine nucleoside is in green and pink, guanosine nucleoside is in pink-only, pink pentagon represents ribose. Yellow asterisks highlight the new emerging linkage.

**5**

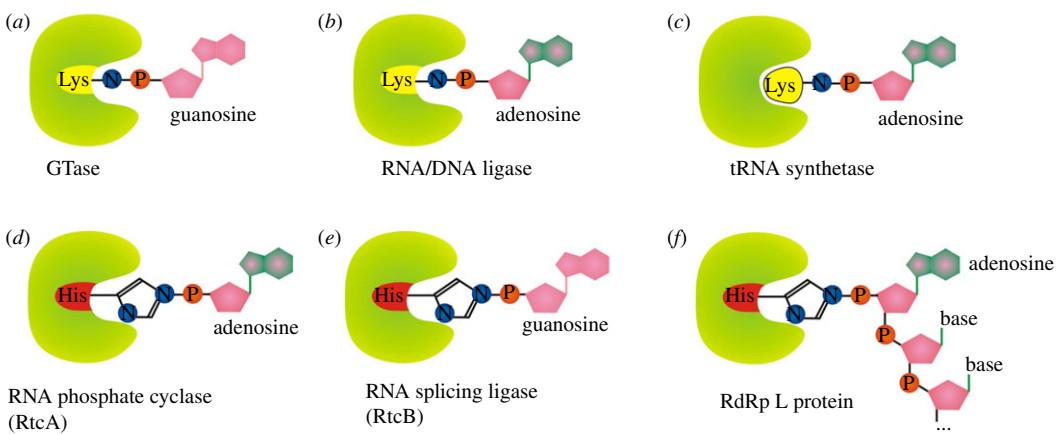

**Figure 2.** Nucleotide-enzyme, RNA-enzyme or nucleotide-amino acid covalently bound intermediates used during 5′ aminoacylation of RNA by different enzymes (3′ aminoacylation by RtcB). (a) RNA guanylyl transferase (GTase), (b) RNA/DNA ligase, (c) tRNA synthetase (LysU). (d) RNA phosphate cyclase (RtcA), (e) RNA splicing ligase (RtcB), (f) RNA-dependent RNA polymerase L protein from VSV.

## 4. Unrelated RNA polymerases are the main, but not the only enzymes installing NC caps

Most of the discovered NC capping is believed to originate from the incorporation of abundant cell metabolites (nucleotide analogues or dinucleotides) to RNA during the initiation step of template-dependent transcription by DNA dependent RNAPs. The nucleotide part of the metabolite pairs with the transcription start site of template DNA forming the 5′ end of nascent RNA.

The first argument for NC *ab initio* incorporation by RNAP was the simple observation that NAD$^+$ incorporated only in RNA, which typically initiates with ATP. Additionally, there is a high correlation between the promoter sequence and the efficiency of NAD$^+$ incorporation [17,22–24], supporting the template dependency of this process. Importantly, NC caps were found only on non-processed RNAs initially—in yeast NAD-RNA on pre-mRNAs, and on mitochondrial transcripts that are not 5′ end processed [25]. Another argument for RNAP II-dependent NAD$^+$ capping is the utilization of different transcription start sites for NAD versus m$^7$G-capped RNAs of the same gene observed in *Arabidopsis thaliana* and *Saccharomyces cerevisiae* [22,26].

Initiation with NC nucleosides seems to be a common feature of structurally and evolutionary unrelated families of transcriptases: (i) multi-subunit RNAPs—bacterial RNAPs and eukaryotic RNAP II, (ii) single-subunit RNAPs—mitochondrial and bacteriophage T7 RNAP [12,13,27–29]. Recently, a replicative enzyme, primase, which makes short RNA primers to be extended by DNA polymerase was shown to cap RNA with NAD$^+$ and FAD *in vitro* [29]. Capping seems ubiquitous, yet there is a conspicuous outlier—a chloroplast. Up until now, only two chloroplast RNAs were found to be NAD-capped [18,26]. One of them was rRNA, which suggests that the capping could be done by one of the phage-type single-subunit polymerase transcribing *rrn* operon of *Arabidopsis* chloroplast [30]. This leaves the question whether the main plastid RNAP of bacterial type is capable of NAD$^+$/NADH incorporation. We tested cyanobacterial RNAPs, the closest bacterial relatives of chloroplast RNAPs (of *Synechococcus elongatus* PCC 7942 and *Synechocystis* sp. PCC 6803) and found that they can efficiently incorporate NAD$^+$/NADH into an RNA (C.J. 2020, unpublished data). We used conditions similar to those published for *E. coli* RNAP [13] for assessing formation of the two nucleotides-long RNA (0.5 mM cofactor and 50 µM CTP as next nucleotide, linear DNA fragment containing RNA I promoter in 20 mM Tris–HCl (pH 7.9), 40 mM KCl, 10 mM MgCl$_2$). Whether the relatively low *in vivo* NAD$^+$/NADH concentration in chloroplast [31] prevents its incorporation as initiating NC cap, or the capped RNA is processed very fast remains to be answered in prospective research.

Despite all these arguments for *ab initio* capping by RNAP, there are indications for other post-transcriptional mechanisms to obtain 5′-NC-capped RNA. The most convincing one is in mammalian cells, where intronic small nucleolar RNAs (snoRNAs) and small Cajal body RNAs (scaRNAs) were reported to be NAD-capped [32]. While the processed intronic RNAs are inherently monophosphorylated [33–35] it can be deduced that 5′ NAD-caps can be added independently of RNAP

action. This review focuses on known and potential alternative pathways of NC RNA capping with abundant AMP analogues.

# 5. RNA polymerases cap RNA with NAD$^+$/NADH: could enzymes of NAD$^+$ biosynthesis pathway cap RNA post-transcriptionally?

Nicotinamide dinucleotide (NAD) is a cofactor required for many cellular oxidases and reductases. As a redox coenzyme, it shuttles between the oxidized form (NAD$^+$) and the reduced form (NADH). However, it has been uncovered in the last few years that the role of NAD$^+$ in cells is much broader. Besides being a redox coenzyme, NAD$^+$ is consumed and used as a co-substrate in the enzymatic reactions by several types of enzymes (e.g. NAD-dependent DNA ligases, poly (ADP-ribose) polymerases (PARPs), ADP-ribosyl cyclases, ADP-ribosyltransferases that modify proteins and small molecules) [36] and excitingly it was found as a 5′ cap of RNAs from bacteria as well as eukaryotes [8,11–13].

NADylated RNA (NAD-RNA) was one of the first discovered modifications of the 5′ end of bacterial RNA. Although expected to originate mainly from transcription initiation with NAD$^+$ (figure 3a) [12], it was also demonstrated to appear on mammalian processed RNA species, providing the proof of the existence of post-transcriptional NAD-capping process [32]. Post-transcriptional NAD$^+$ capping may result from at least three different events (figure 3b): (i) covalent binding of the whole NAD$^+$ to any nucleoside triphosphate on the 5′ end of RNA, (ii) transfer of nicotinamide ribonucleotide to 5′ ATP-terminated RNA, or (iii) transfer of nicotinamide to ADP ribosylated RNA (ADPR-RNA). So far, no post-transcriptionally NAD-capping enzyme has been described, and identification of such an enzyme would provide a useful tool for studying NAD capping function and regulation.

A suitable post-transcriptional NAD-capping candidate may be recruited from nicotinamide mononucleotide adenylyltransferases (NMNATs) or other enzymes of the NAD$^+$ biosynthesis pathway. NMNATs catalyse the transfer of the AMP moiety of ATP to nicotinamide mononucleotide (NMN) to form NAD$^+$ (figure 3c), and they also catalyse the reverse reaction (the generation of ATP and NMN from NAD$^+$ or alternative dinucleotides and pyrophosphate [37]). We speculate that the analogous activity of NMNAT might be useful for capping of 5′ATP-terminated RNA with a nicotinamide ribonucleotide/nicotinic acid mononucleotide moiety (figure 3b(ii)) and potentially also for decapping of NADylated RNA.

It was previously shown that the enzymatic substrate-binding pocket of some NMNATs is not strictly specific to ATP, but it can also adopt phosphorylated proteins [38] or different nucleotides and dinucleotides (e.g. reduced or phosphorylated forms of NAD (NADH, NADP+, respectively) [39], nicotinic acid adenine dinucleotide (NAAD), nicotinamide guanine dinucleotide (NGD) [37]). These findings indicate plasticity of the active site of at least some NMNATs, permitting speculations about possible RNA capping abilities of enzymes from this family.

Cells usually code for more isoforms of NMNATs (two in bacteria [40,41], two in yeast [42], three in human cells [37]). Due to different subcellular localization, tissue specificity of particular isoforms and their co-expression, they are expected to carry on defined, non-redundant functions. However, their specific functions are still enigmatic. Human NMNAT1 is a nuclear protein, NMNAT2 and NMNAT3 are localized to the Golgi complex and the mitochondria, respectively. Mitochondrial localization of NMNAT3 makes it an interesting target for further testing of its potential capping activity since 40–70% of NAD$^+$ in cells resides in the mitochondria [43,44] and up to 15% of human mitochondrial RNA and up to 60% of yeast mitochondrial RNA is NAD-capped (although in yeast mitochondria, strictly unprocessed transcripts were found to be NAD$^+$-capped [25]). Moreover, kinetic properties showed that particularly NMNAT3 exhibits a high tolerance towards substrate modifications and it can convert NAAD, NGD and NADH to the same extent as NAD$^+$ [37].

Two NMNATs with different substrate specificities were identified in *E. coli*: NadD (YbeN) which preferentially uses nicotinic acid mononucleotide (NaMN; 20 times more efficiently than NMN) [41,45,46] and NadR, which prefers NMN (170 times more efficiently than NaMN) [40]. Cyanobacterial NMNAT (slr0787 or NadM_SYNY3) also harbours a highly conserved sequence that classifies it as NUDIX hydrolase (hydrolases cleaving nucleoside diphosphates linked to moiety X) specific for ADP-ribose and 2′-phospho-ADP-ribose [47].

To test a hypothesis of NMNATs' ability to cap RNA with NAD$^+$ we overproduced and purified recombinant NadD and NadR proteins from *E. coli.* We tested their potential capping and decapping activities *in vitro* by incubating them with radiolabelled ppp-RNA or NAD-RNA in conditions

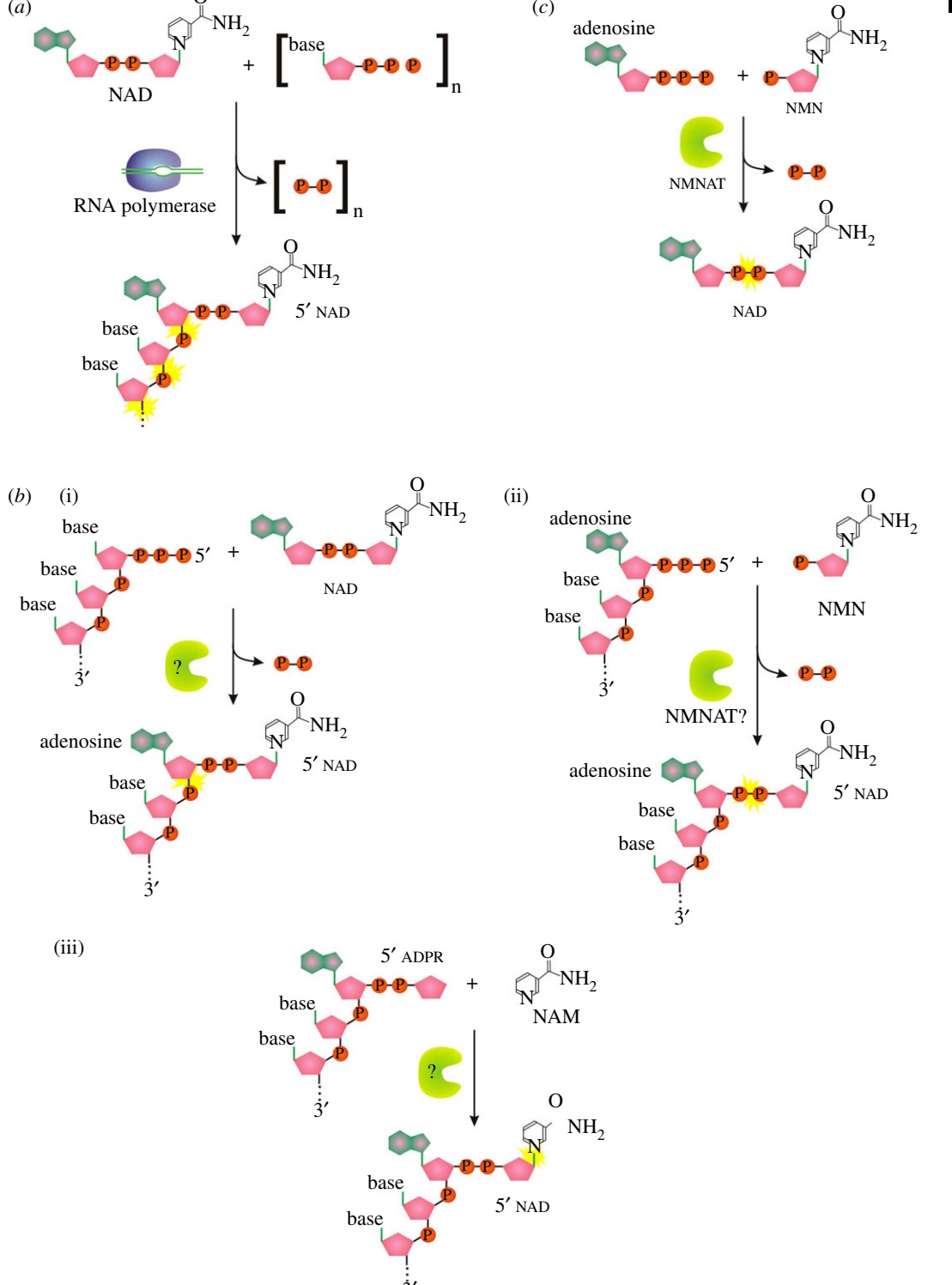

**Figure 3.** (*a*) Mechanism of *ab initio* NAD capping by RNAP. (*b*) Three possible pathways to produce NAD-RNA post-transcriptionally. (*c*) Known function of NMNATs in NAD synthesis. Symbols as described in figure 1.

suitable for NAD synthesis as specified by Rafaelli *et al.* for NadR (50 mM Hepes, pH 8.6, 10 mM MgCl$_2$, 1 mM NMN) [40] and by Mehl *et al.* for NadD [41] (100 mM Tris–HCl, pH 8.0, with 2 mM MgCl$_2$, 5 mM NMN) but did not observe either capping or decapping activity (J.W. 2020, unpublished data). The absence of activity of NadR and NadD towards RNA may reflect the absence of such a mechanism in bacteria, or alternatively the need for condition optimization or auxiliary factors. NAD$^+$

post-transcriptional capping can also be performed by another unknown enzyme. All things considered, a test of the involvement of eukaryotic NMNAT in NC RNA capping is still worthwhile.

# 6. Capping RNA with dinucleotide polyphosphates (Np$_n$Ns) by RNA polymerase and aminoacyl tRNA synthetase

During the last 2 years, the 5′ RNA cap family was expanded by a number of new caps formed by dinucleotide polyphosphates [10,14,21,48]. For more than five decades, dinucleoside polyphosphates (Np$_n$Ns) were known as signalling molecules in all life domains, but their target and physiological function were elusive [49,50]. Various Np$_3$Ns and Np$_4$Ns are being synthesized by a side reaction during aminoacyl-tRNA synthesis by lysyl-tRNA synthetase (LysU, figure 4b, reactions (i) and (ii) versus (i) and (iii)) [50–52]. But the range of producers is possibly wider while overproduction of other aminoacyl -tRNA synthetases (methionyl- phenylalanyl- or valyl-) affects the Np$_n$N levels in *E. coli* as well [51,53]. In bacteria under stress conditions like oxidative stress or heat shock, the concentration of Np$_n$Ns can increase from the µM to the mM range [54–56], which is why they are thought to act as alarmones [53]. In eukaryotes, they play a role in neuronal signalling, immune response and cardiovascular function [57].

Papers from two independent groups demonstrated that RNA can be capped *in vivo* by a number of different Np$_n$Ns and their methylated forms: Ap$_3$A, m$^6$Ap$_3$A, Ap$_3$G, m$^7$Gp$_4$Gm, Ap$_5$A, m$^6$Ap$_5$G, m$^6$Ap$_4$G, m$^6$Ap$_5$A and 2mAp$_5$G [10,14]. The type of capping is specific for the growth phase and can be induced by disulphide stress. All presented Np$_n$N studies are consistent with the hypothesis that Np$_n$N capping may be the signal-transfer pathway that mediates transmission of Np$_n$N levels to bacterial gene expression by changing the lifetime of capped RNAs. Lifetime is affected by decapping enzymes (primarily ApaH), which are specifically inhibited by inducers of disulfide stress, stabilizing its target RNAs [10,14,21]. This statement is consistent with findings of previous work that ApaH mutation causes Ap$_4$A accumulation in *E. coli* [58]. The Np$_n$N-RNA is further stabilized by methylation of the cap, inhibiting RppH (RNA pyrophosphohydrolase)-dependent degradation. This is probably caused by introducing the positive charge to the purine ring by the methylation followed by loss of the interaction with two RppH arginines responsible for the purines binding [10]. The process of *in vivo* methylation of Np$_n$N caps needs to be deciphered in future studies.

The Belasco group demonstrated that RNAP is expected to be the enzyme responsible for Np$_n$N capping of RNA *in vivo* (figure 4a). They manipulated the promoter sequence upstream of the site of transcription initiation and showed that it influences both Np$_4$A incorporation into nascent transcripts *in vitro* and levels of Np$_4$ capping *in vivo* [14]. The ninefold preference of RNAP to initiate transcription with Np$_n$N rather than ATP also supports the RNAP-dependent origin of most Np$_n$N-RNA in the cell [21]; however, it was not observed by the Cahová group [10] (the authors observed at least two times better incorporation of ATP compared with Ap$_6$A or Ap$_4$G, similarly to NAD$^+$ incorporation).

Interestingly, Ap$_4$N-RNA can be produced *in vitro* by lysyl-tRNA synthetase LysU in the presence of triphosphorylated RNA, ATP and lysine (figure 4b, reactions (i) and (iv)). A supplementary role for aminoacyl-tRNA synthetases or other cellular enzymes in capping ppp-RNA by nucleotidyl transfer to the RNA 5′ end cannot be excluded; however, the contribution is likely to be relatively small [14]. All enzymatic reactions carried by LysU (and probably also by other aminoacyl tRNA synthetases [53]) require the amino acid-NTP covalent intermediate (figure 2c and 4b) which draws a parallel to other nucleotidyl transferases known to cap RNA *in vitro* or *in vivo*. In contrast to other discussed mechanisms in this review, the amino acid is not covalently bound to the capping enzyme but is tightly bound in the active centre of the enzyme.

Concurrently with Np$_4$N level rise during heat shock and oxidative stress, relevant RNA capping [14] is expected to rise. In connection with this notion, it is interesting that heat shock and oxidative stress proteins DnaK, GroEL, E89, C45 and C40 in *E. coli* bind Ap$_4$A [59]. It would be interesting to test whether Ap$_4$A-RNA can interact with these enzymes and localize them to the site of RNA translation. Particularly chaperone activity of DnaK/GroEL localized at the site of emerging nascent protein during heat shock could be beneficial for its correct folding which can be compromised by high temperature.

Decapping of RNA by ApaH may be a more widespread phenomenon than previously expected. ApaH-like phosphatase (TbALPH1) is the major mRNA decapping enzyme of trypanosomes where no clear orthologues to Dcp2 (the main decapping enzyme in most studied eukaryotes) or its

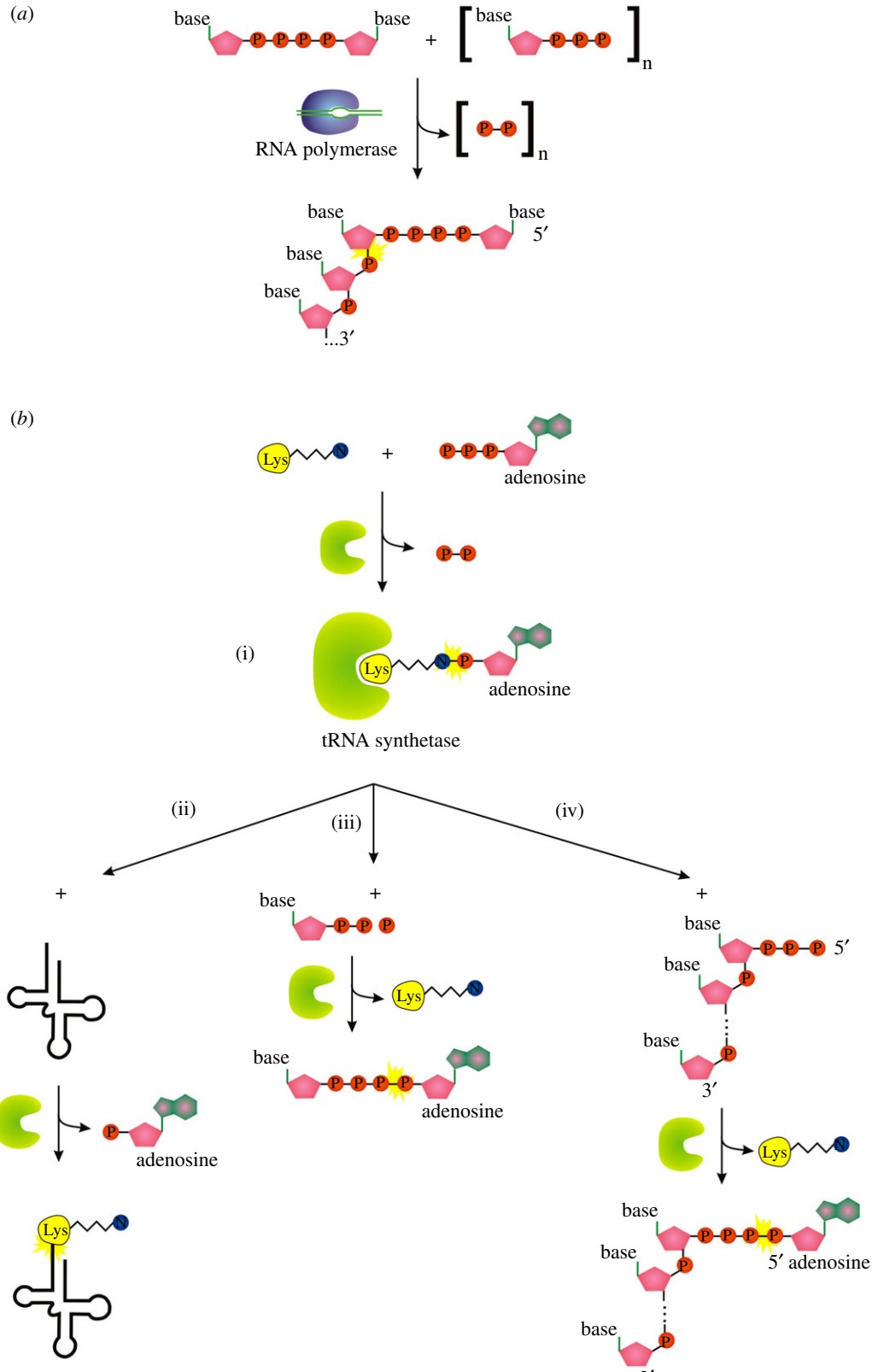

**Figure 4.** Pathways capping RNA with Np$_4$N cap. (*a*) RNAP *ab initio* capping. (*b*) Aminoacyl tRNA synthetase pathway mediating nucleotidyl transfer dependent on amino acid-AMP covalent intermediate. (i) Formation of an amino acid-AMP intermediate, (ii) charging of tRNA, (iii) production of Np$_4$A alarmone, (iv) Ap$_4$N capping of RNA. Symbols as described in figure 1.

associated proteins are present. While the genome of trypanosomes is transcribed as long, polycistronic pre-mRNAs of up to 100 protein-coding genes and lacks conventional promoters, the missing transcriptional regulation in these parasites must be supplemented by post-transcriptional mechanisms,

including the regulation of mRNA decay. RNAs of trypanosomes (and other Kinetoplastida species) have a unique, heavily methylated cap4 structure which can be removed by TbALPH [60]. Similarly, methylated $Np_4Ns$ of *E. coli* are removed by ApaH [10], whereas bacterial RppH decapping is fully inhibited by the cap methylation. This parallel may be useful for studying ApaH-like recognition and decapping mechanisms of methylated cap structures in higher organisms and may shed light on the evolution of similar structures.

# 7. ADP-ribose is a potential new major non-canonical RNA 5′ cap with at least four possible mechanisms for capping

ADP-ribosylation is a reversible chemical modification catalysed by ADP-ribosyltransferases called poly (ADP-ribose) polymerases (PARPs). The essence of the process is a transfer of monomers or polymers of ADP-ribose nucleotide from $NAD^+$ cofactor onto macromolecular targets such as proteins and nucleic acids (see the ADPR transfer to RNA, figure 5*b*). ADP-ribosylation affects a wide range of key biological processes: DNA-damage repair, DNA replication, transcription, cell division, signal transduction, stress and infection responses, microbial pathogenicity and ageing [61].

The canonical function of 2′-phosphotransferase (Tpt1), a member of PARP family, is the $NAD^+$-dependent conversion of an internal RNA 2′-phosphate to a 2′-OH using ADPR-RNA intermediate (figure 5*b*) [36]. Intriguingly, despite Tpt1 homologs being found in all domains of life, 2′-phosphate tRNAs are normally produced during tRNA splicing only in fungi. Other organisms use different splicing mechanisms [36]. Therefore, substrates and the biological functions of the Tpt1 homologs in these species were enigmatic. Recently, a subset of Tpt1 enzymes (bacterial and archaeal Tpt1, and other members from PARP family from eukaryotes: human PARP10, PARP11, PARP15, TRPT1) were demonstrated to catalyse $NAD^+$-dependent ADP-ribosylation of an RNA or DNA 5′-monophosphate terminus *in vitro*, producing ADPR-RNA (figure 5*c*) [62,63].

Although ADPR-RNAs are not yet found *in vivo*, the conditions used during *in vitro* experiments strongly suggest that the same reactions may occur in cells [62,63]. Another hint is the existence of ADP-ribosylhydrolases, able to decap ADPR-RNA [62]. Moreover, the existence of analogous 5′ ribosylated DNA was already predicted by experiments in cell-free extracts and preliminary *in vivo* data [64–66]. ADPR-RNA can be alternatively attached to the RNA 5′ end during transcription initiation by RNAP (figure 5*a*) but ADPR concentration in the cell is three orders of magnitude lower than ATP or $NAD^+$—$4.3 \times 10^{-6}$ versus $9.6 \times 10^{-3}$ and $2.6 \times 10^{-3}$ M, respectively [67], which makes the process of *ab initio* capping by RNAP less plausible but does not exclude its usage by RNAP as initiating substrates in a specific situation.

As recently discovered, ADPR-RNA can be also produced by hydrolysis of N-glycosidic bond of NAD-RNA by eukaryotic glycohydrolase CD38 *in vitro* (figure 5*d*) [68].

If found on RNA *in vivo*, ADP-ribose would be a new NC RNA cap, possibly present in all domains of life. It could be part of anti-viral immunity as ADP-ribosylation-dependent response of a host, which could be counteracted by virus-induced decapping of ADP-ribosylated RNAs [62]. This hypothesis is supported by a recent discovery that a conserved macrodomain-containing nsP3 protein of many viruses (including SARS-Cov2, the cause of COVID-19) hydrolyses ADP-ribosylated RNAs [69].

# 8. Potential capping enzymes producing A-pp-RNA: RNA ligases, RNA 3′ phosphate cyclases and tRNA guanylyl transferases

## 8.1. RNA ligases

ATP- and NAD-dependent DNA ligases and ATP-dependent RNA ligases comprise a superfamily together with the above-mentioned GTP-dependent mRNA capping enzymes (guanylyl transferases, figures 1*a* and 2*a*). Similarly to GTase, DNA/RNA ligases catalyse 3-step nucleotidyl transfer to polynucleotide 5′ ends via covalent enzyme-(lysyl-N)-AMP intermediates (figure 2*b*) [70]. In the first step, attack on the α-phosphorus of ATP or $NAD^+$ by the enzyme results in the release of pyrophosphate or nicotinamide mononucleotide (NMN), respectively, and formation of a covalent enzyme-(lysyl-N)–NMP intermediate (figure 6*a* reaction (i)). In the next step, the AMP (*p*-A) is transferred to the 5′ end of p-DNA/p-RNA forming A-pp-DNA/A-pp-RNA (ii). In the last step,

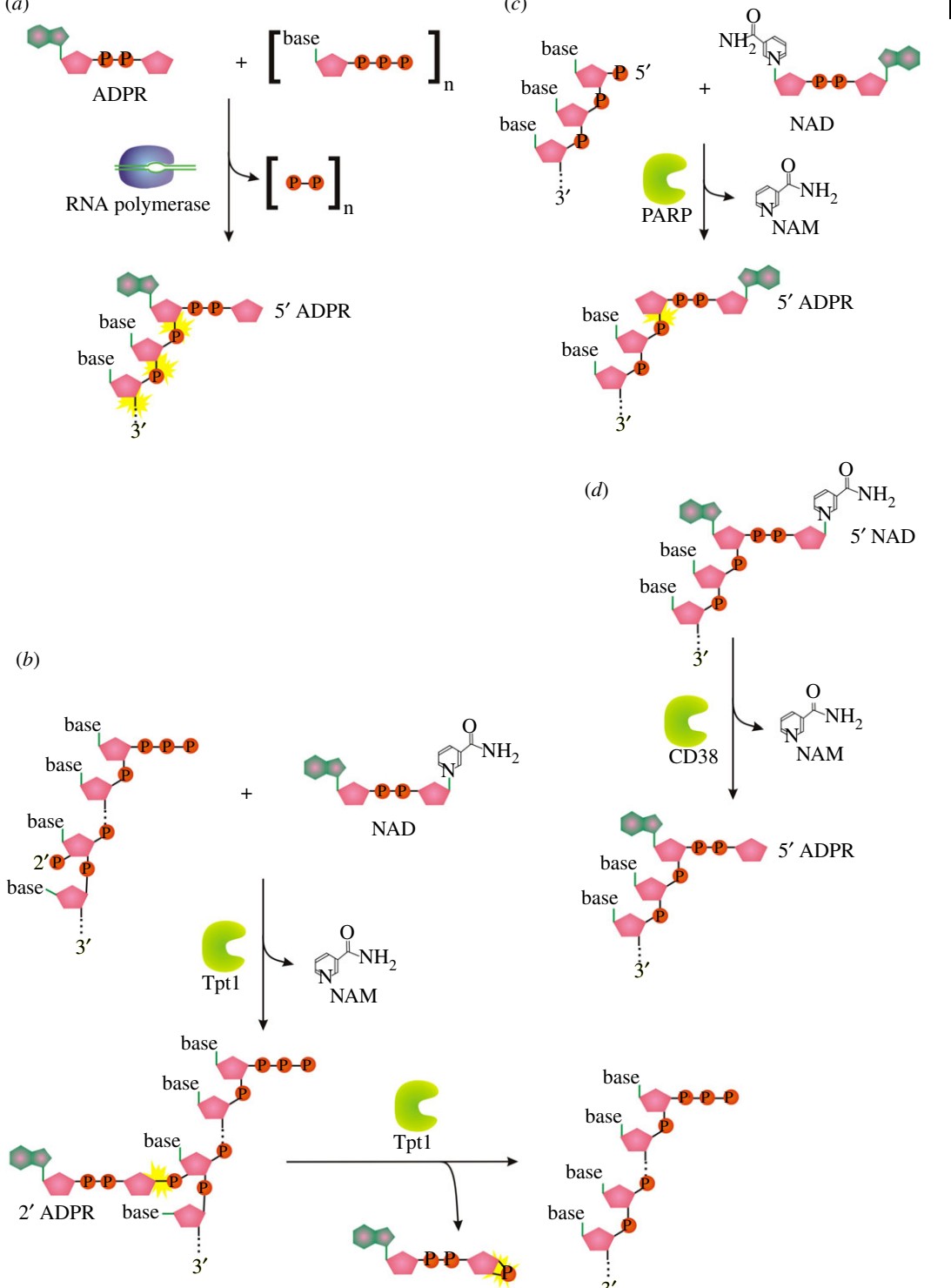

**Figure 5.** (*a*) Proposed mechanism for 5′ RNA capping with ADPR by transcription initiation by RNA polymerase. (*b*) Tpt1-mediated transfer of an internal RNA 2′-monophosphate (2′ p) to NAD+ to form a 2′- OH RNA, ADP-ribose 1″,2″ cyclic phosphate, and nicotinamide [36]. (*c*) Proposed reaction mechanism for Tpt1-dependent 5′ RNA capping with ADPR. (*d*) Mechanism of decapping of NADylated RNA by ADP ribosyl cyclase (CD38) producing ADPR-capped RNA. Symbols as described in figure 1.

DNA/RNA ligases catalyse attack by the 3′-OH of the nick on the A-pp-DNA/RNA to join two polynucleotides and release AMP (iii). At least in the case of DNA ligases, the final ligation step has variable efficiency, therefore releasing high levels of adenylated nucleic acid intermediate [71], the efficiency of RNA ligases potentially releasing adenylated RNA is unknown.

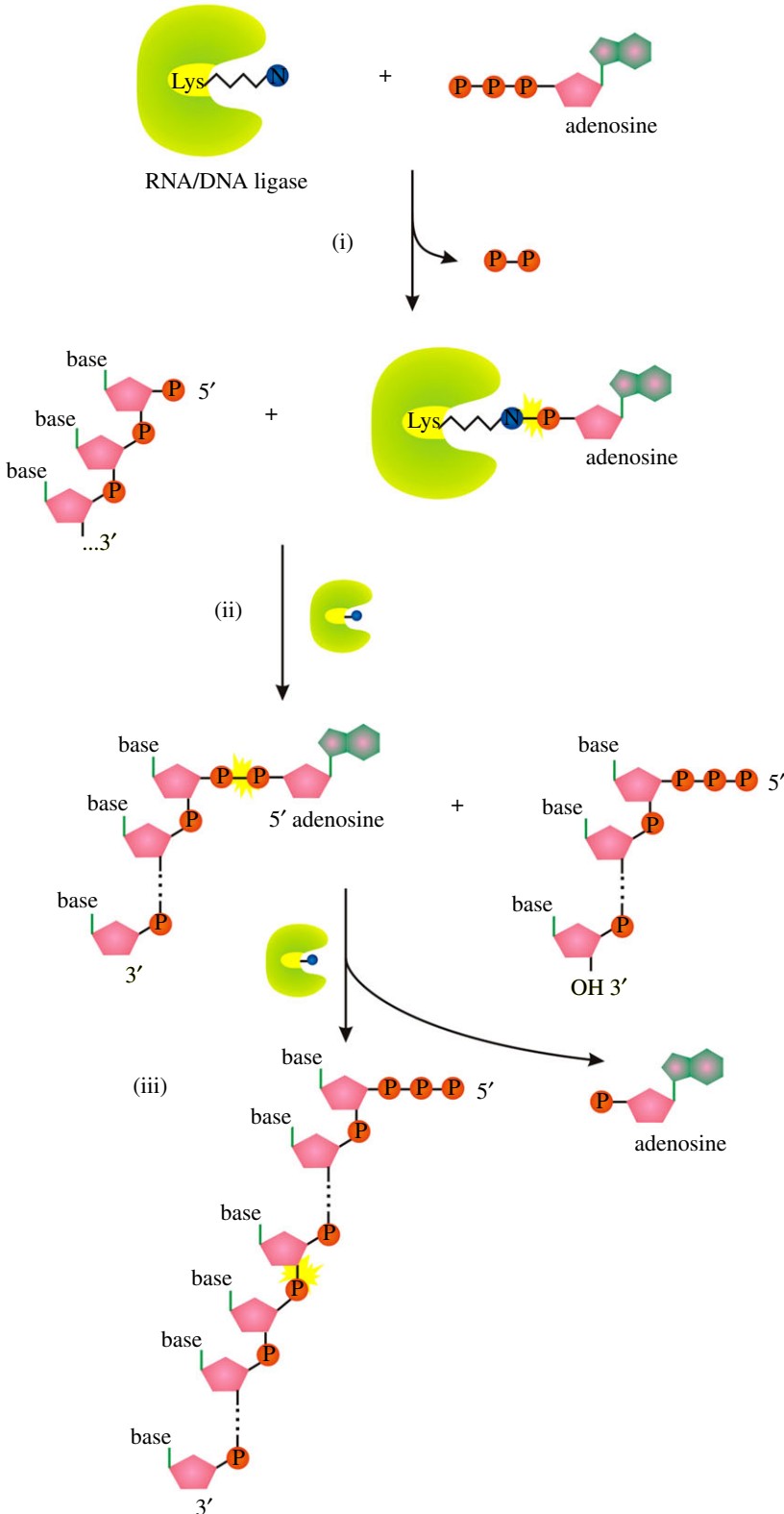

**Figure 6.** Mechanism of RNA ligation by ATP-dependent RNA ligases. Symbols as described in figure 1.

## 8.2. RNA 3′-phosphate cyclase RtcA

A similar ligase-like mechanism of 5′-DNA/RNA adenylylation is catalysed by the RNA 3′-phosphate cyclase (RtcA) [72]. *Escherichia coli* RtcA catalyses adenylylation of 5′-p ends of DNA or RNA strands using RtcA-histidinyl-N-AMP intermediate to form A-pp-DNA and A-pp-RNA products (figure 7a).

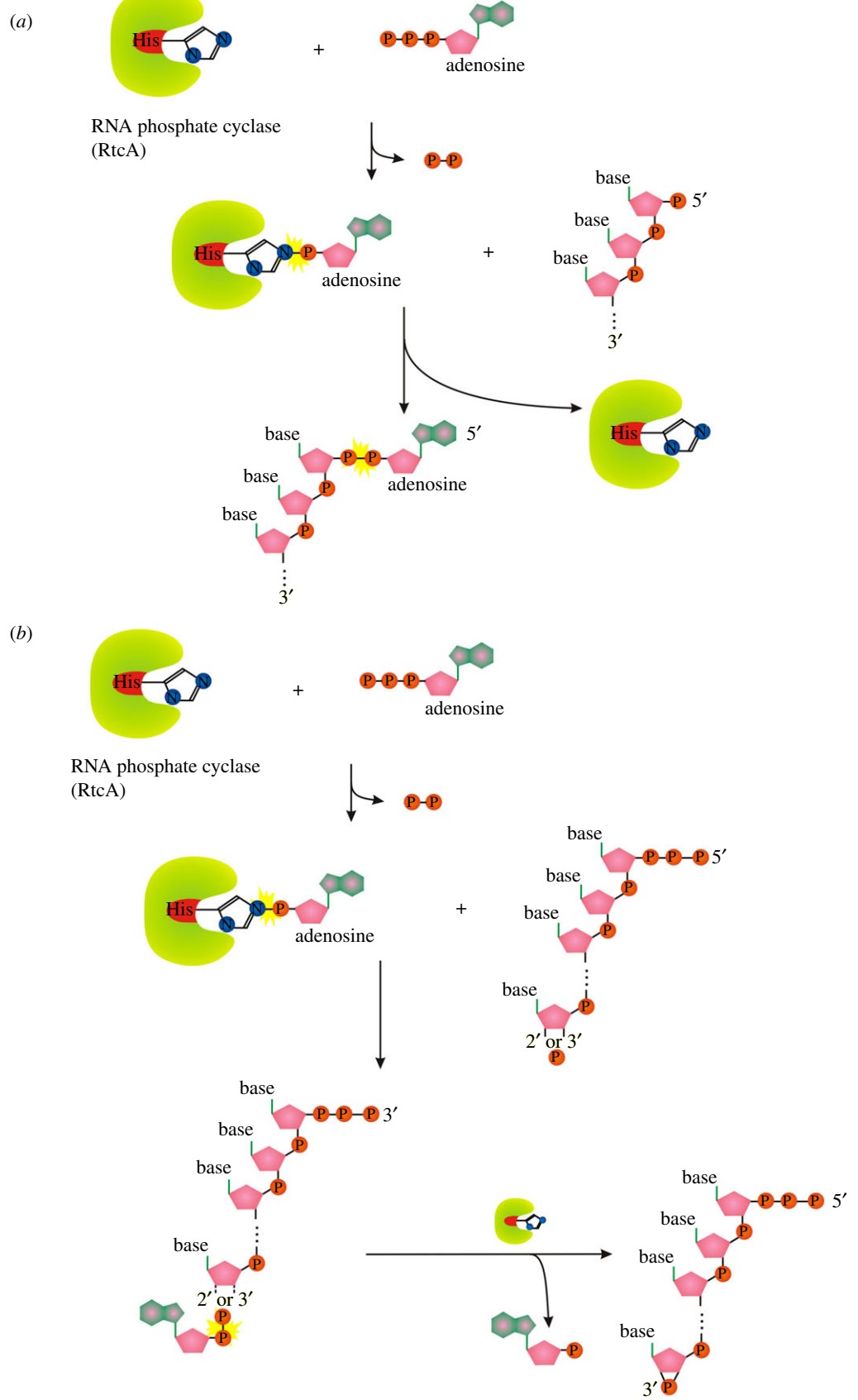

**Figure 7.** (*a*) Mechanism of 5′ RNA/DNA adenylylation by RtcA. (*b*) Mechanism of RtcA 2′,3′ phosphate cyclization. Symbols as described in figure 1.

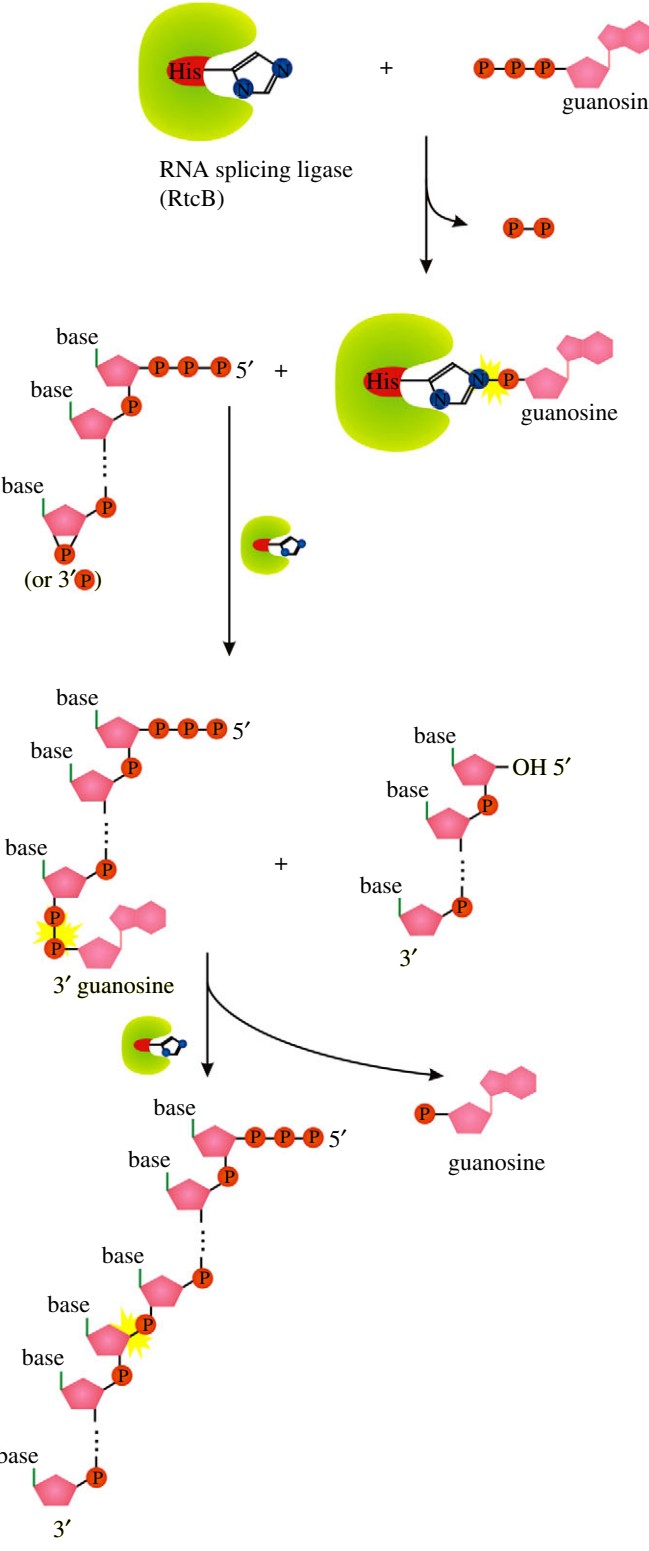

**Figure 8.** Splicing mechanism of RtcB. Symbols as described in figure 1.

The canonical function of RtcA is a catalysis of the ATP-dependent conversion of a 3′-p-RNA or 2′-p-RNA end to a 2′,3′-cyclic phosphate via covalent enzyme-histidinyl-N-AMP and RNA-3′pp-A intermediates (figure 7b) [73]. RtcA is coregulated in an operon with an RNA ligase, RtcB, that splices RNA 5′-OH ends to either 3′-phosphate or 2′,3′-cyclic phosphate ends (figure 8) [74]. RtcA might serve an end healing function in an RNA repair pathway, by converting RNA 2′-phosphates, which cannot be spliced by RtcB, to 2′,3′-cyclic phosphates that can be sealed.

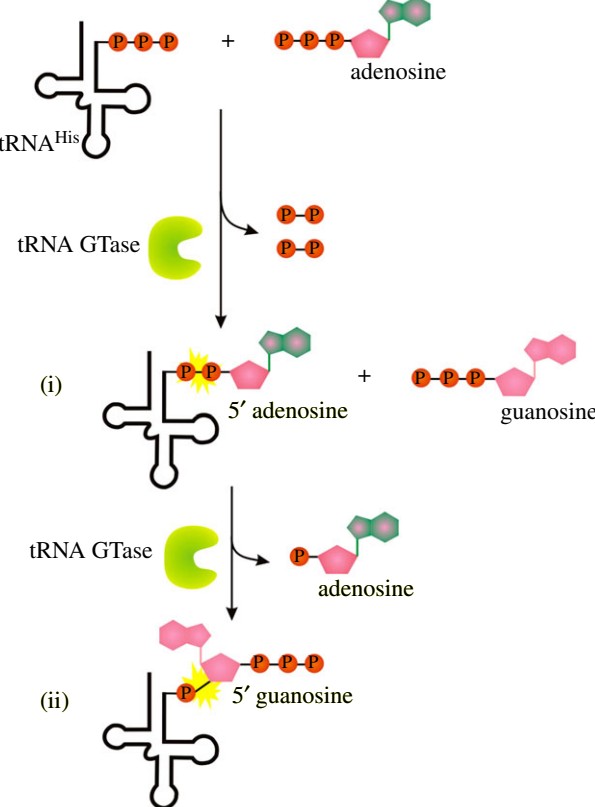

**Figure 9.** Mechanism of 3′-5′ GTP (ppp-G) addition by tRNA guanylyl transferases. Symbols as described in figure 1.

The main difference among the conventional DNA/RNA ligases, RtcA and RtcB is the amino acid used to form the reaction intermediate (RtcA with ATP/RtcB with GTP form a covalent RtcA-histidinyl-N-AMP/RtcB-histidinyl-N-GTP intermediate, ligases use enzyme-lysyl-N-AMP intermediate; figure 2b,d,e). RtcB and RtcA are structurally unrelated proteins with no similarities in their active sites [75–77]. They also lack structural similarity with RNA/DNA ligases, either globally or with respect to their active site architectures [75,78].

The ability of *E. coli* RtcA to catalyse adenylylation of 5′p ends of DNA or RNA strands to form A-pp-DNA and A-pp-RNA products opens the possibility of new cellular functions in RNA capping and therefore affecting the properties of capped RNAs. Such possibilities need to be tested.

## 8.3. tRNA guanylyl transferases

5′-5′ mRNA GTases, and therefore also canonical caps, are unique for eukaryotes and missing in prokaryotes and archaea. Interestingly, a similar enzyme (tRNA guanylyl transferases; tRNA GTase), is present in all three domains of life (THG1 in human, Thg1 in yeast and Thg1-like proteins (TLPs) in archaea and bacteria) [79,80]. tRNA GTase is the only known example of an enzyme that catalyses templated nucleotide addition in the 3′-5′ direction, opposite to that of all known DNA and RNAPs [81] resulting in ppp-G-RNA (compare with G-ppp-RNA produced by mRNA GTase). In the first step, it uses ATP to activate the monophosphorylated tRNA$^{His}$ (p-tRNA$^{His}$ generated by ribonuclease P) producing adenylated tRNA$^{His}$ (A-pp-tRNA$^{His}$) intermediate (figure 9(i)). In the second step, the hydroxyl of a GTP nucleotide attacks the activated intermediate, yielding the triphosphorylated form of tRNA$^{His}$ (ppp-G-tRNA$^{His}$) and AMP (figure 9(ii)) [80–82].

The adenylylated intermediate product of the pathway (A-pp-tRNA) can be classified as NC-capped RNA but while the process of adenylylation and subsequent guanylyl transfer is specifically bound to recognition of anticodon of tRNA$^{His}$ [79], its wider use in RNA NC capping is questionable.

## 9. FAD capping by RNA polymerase and potentially flavin biosynthesis enzymes

Flavin adenine dinucleotide is one of the cellular metabolites reported as covalently bound to the RNA in prokaryotes, eukaryotes and viruses [7]. It is readily incorporated by cellular and mitochondrial RNAPs to 5′ end of the nascent transcript [13,27]. FAD as well as NADH serves as NC initiating substrates for a primase and affect replication primer processing *in vitro* [29].

Whether there are other enzymes producing FAD-RNA remains an unanswered question. We speculate that enzymes from the flavin mononucleotide adenylyl transferase (FMNAT) family could be a possible candidate, analogically to NMNATs as candidates for $NAD^+$ capping.

## 10. Non-canonical RNA capping by ribozymes

Intriguingly, not only proteins can cap RNA with metabolite cofactors. Ribozymes able to make $NAD^+$-, FAD- and CoA-capped RNA by attaching small-molecule precursors to a 5′-terminal ATP were evolved *in vitro* [83]. Ribozymes are generally believed to be functional fossils from RNA world [84]. Hypothetically, the ribozyme reactions might once have been the main pathway, and could also today still be working unrecognized in *in vivo* synthesis of capped RNA [85].

## 11. Protein caps of the viral genome

Some positive sense single-stranded RNA viruses have viral protein genome-linked (VPgs) covalently attached to the 5′ end of their genome. In caliciviruses, VPgs were confirmed to interact with the cap-binding protein (eIF4E) and to be essential for translation [86]. These viral cap-like structures are not small molecules but proteins, and thus can be classified as NC 5′ RNA caps.

## 12. The extent of NC capping and cellular roles of capped RNAs are still unclear

A number of potential physiological roles of NC capping are suggested in two recent reviews, including RNA folding and stability, cellular localization, etc. [85,87]. Yet only a few actual examples are published, e.g. increased NADylation of RNAIII, a central quorum-sensing regulator of *Staphylococcus aureus* repressed production of toxins, thus decreasing cytotoxicity of the bacterium [88]. Therefore, so far, the extent, impact or functions of NC-capped RNAs remain largely unknown. To demonstrate this point, we would like to highlight a few enigmatic and still controversial aspects of NAD capping, the most studied NC capping to date, thanks to the number of NAD-RNA isolation and quantification techniques [11,24,26].

It is unclear if a majority of the capped transcript are full-sized or truncated species. In bacteria (*E. coli*, *B. subtilis* and *Staphylococcus aureus*), the abundant NAD-capped species are 5′-terminal fragments of certain mRNAs [11]. Fragmented NAD-RNAs were also found in *B. subtilis* dormant spores [89]. Nevertheless, validation of some mRNA species detected also full-length transcripts in *B. subtilis* and *S. aureus* [17,88]. These discrepancies may be caused by the bias of the original method, where only short RNAs (less than 200 nt) were used for NGS sequencing library preparation [11]. Indeed in yeast, mammals and plants, the observed NAD-RNAs were mostly of the full length [18,25,26,32]. However, recent preprint publication from yeast reports yet again a large number of short mRNA 5′-terminal fragments [22], echoing findings in bacteria. The important question to answer is how these short $NAD^+$-capped RNAs are produced, whether the presence of $NAD^+$ at the 5′-end affects either transcription initiation, termination or degradation of full-length RNAs to make species of shorter yet defined length, stable enough to be detected, or whether the short length is just an artefact of NAD-RNA identification process.

The impact of capping on transcription and post-transcriptional processing of RNA is known only for a few examples. In *S. cerevisiae* and *A. thaliana*, RNAP II uses different transcription start sites (TSS) for NAD-RNA and for $m^7G$-capped RNAs of the same gene, but the biological function of alternative TSSs remains unsolved. It was shown for *E. coli* transcription that 5′-NAD can stabilize short transcript and prevent their release improving chances for RNAP to escape into a productive elongation [13].

Mammalian and plant NAD-RNAs can be spliced and polyadenylated [18,32], suggesting they are treated at least by some of the cellular machinery as canonical mRNAs.

The translatability of NC-capped mRNA is a conundrum. Reports on the translatability of NADylated mRNAs are conflicting. While in human and yeast cells NAD-RNAs do not support translation but instead promote mRNA decay [22,32], NAD-RNA species of plants are enriched in the polysomal fraction and associate with translating ribosomes, suggesting that they are translated [18]. Reports from bacteria about NAD-RNA translation are scarce. In *S. aureus* 5′-NAD cap on RNAIII might impair translation of a small open reading frame inside this regulatory RNA *in vivo*, but the impact of additional factors cannot be ruled out. Both NAD⁺-RNAIII and ppp-RNAIII supported the formation of stable translational initiation complexes *in vitro* [88]. In summary, utilization of NAD-mRNAs for its main purpose (translation) is probably ceased either by the truncation of the resulting NAD-RNA by an unknown mechanism (mostly in bacteria) or by its inability to be translated (in human and yeast cells). The translatability is feasible in mitochondria, where NAD⁺ capping seems to be higher than in other cell compartments or prokaryotic cells (up to 15% in human and 60% in yeast [19]). It is hard to imagine that so high number of mitochondrial transcripts would be made just for being degraded, so NAD⁺ capping of mitochondrial RNAs seems to be at least tolerated if not required for efficient translation.

# 13. Summary and perspectives

In this review, we compiled information about a number of enzymes beyond RNAPs capable of *in vitro* or *in vivo* addition of NC caps to RNA. All the reviewed enzymes with proven capping activity decorate RNA by 5′-5′ addition of nucleoside connected by a variable number of phosphates or phosphate-ribose-phosphate bridges. Interestingly, most of these enzymes share the principal mechanism of cap addition with canonical 5′ RNA capping: they use amino acid—nucleotide covalent linkage as an intermediate for nucleotidyl transfer to RNA/nucleotide (figure 2). This common feature can serve as a useful clue for searching other undiscovered capping mechanisms. For example, adenylylation enzymes (AMPylators) using intermediates with covalent phosphodiester bonds between amino acid (serine, tyrosine or threonine) side chain of the enzyme and adenosine monophosphate [90] could be a possible source of capping enzymes. Despite NAD⁺ seeming to be one of the most ubiquitous NC caps, no post-transcriptional capping enzyme was identified for it so far. We tested a possibility of NAD-RNA capping by enzymes of NAD⁺ synthesis pathway (*E. coli* NMNATs NadD and NadR) but did not observe any capping or decapping activity in tested conditions.

Initially, NC capping was perceived as RNAP-dependent stoichiometric event affected by the ratio of concentrations of NTPs to other NTP-analogues. This review focuses on the facts which challenge this perception, showing that cells are probably packed with enzymes capping and decapping RNA with NC caps. Whether all NC capping events are stochastic side reactions of moonlighting enzymes with established primary function, or there exist dedicated machinery for NC capping, remains to be shown. Expanding repertoire of caps and capping enzymes suggests if not a physiological function then at least a major physiological impact of these caps.

Data accessibility. This article has no additional data.

Authors' contributions. This is a review paper with unpublished experimental results discussed. J.W. wrote the manuscript and performed experiments, C.J. performed experiments, Y.Y. wrote and edited the manuscript.

Competing interests. We declare we have no competing interests.

Funding. This work was supported by the Royal Society URF to Y.Y. and the Leverhulme Trust [J.W., Y.Y. grant no. RPG-2018-437].

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
