## [Peer Review File · Royal Society Open Science]

Review History

RSOS-201979.R0 (Original submission)

Review form: Reviewer 1 (Ben Luisi)

Is the manuscript scientifically sound in its present form?

Yes

Are the interpretations and conclusions justified by the results?

Yes

Is the language acceptable?

Yes

Do you have any ethical concerns with this paper?

No

Have you any concerns about statistical analyses in this paper?

No

Recommendation?

Accept with minor revision (please list in comments)

Comments to the Author(s)

This review summarises current state of knowledge on the variety of non-canonical RNA caps discovered to date, the enzymes involved and their mechanisms, both in eukaryotes and in prokaryotes. The authors explain the mechanism of canonical RNA capping and then propose different non-canonical (NC) capping mechanisms. They address NC capping at the initiation of transcription and post-transcription and propose different enzymes that might be involved in these processes. Apart from the widely described NAD⁺/NADH, they also show some of the molecules recently identified as NC including dinucleotide polyphosphates, ADP ribose, and others. Finally, the authors state some of the questions regarding the mechanisms of NC capping and the physiological function of these alternative RNA caps that remain unanswered.

Main comments:

The article can be difficult to read for those who are not familiar with the enzymes and metabolites. Many enzymes are involved: a table can be useful to summarise which enzyme is supposed to be required for a particular non-canonical cap and what is the hypothetical mechanism of the non-canonical capping. There are many abbreviations used, and it might be helpful to also summarise these in the table.

Figures 1 and 3-8 are useful to understand the reactions but could be expanded beyond the simple text presented. Having more graphical figures, with 2D structures of the different interacting molecules and schemas for the protein involved, similar to the ones in Figure 2, might help the reader follow the different mechanisms explained in the Review.

Some positive sense ssRNA viruses have viral protein genome-linked (VPgs) covalently attached to the 5' end of their genome. In caliciviruses, VPgs were confirmed to interact with the cap-binding protein, eIF4E, and to be essential for translation (Goodfellow et al., 2005). Although they are not small molecules but proteins, it might be interesting to mention these viral alternatives as NC 5' RNA caps.

In the second paragraph of page 4 (or page 5 of 17 according to the enumeration at the top of each page). They mention that no NAD-capped RNAs were found in chloroplasts according to Wang et al. (2019), but Zhang et al. (2019) describe a NAD-capped chloroplast rRNA.

Then, they mention an experiment they performed with cyanobacterial RNAPs which could efficiently incorporate NAD⁺ into RNA (same page lines 16-20). It would be helpful if they explained a bit more their new findings (what proteins they used? what conditions? etc.) as they do with their experiments with NMNATs NadD and NadR in pages 5-6 (pages 6 of 17 and 7 of 17). (Wang et al., 2019) suggest the absence of NAD-capped RNA in chloroplasts could be due to low levels of free NAD⁺ in this organelle. When they assessed if cyanobacterial RNAPs could incorporate NAD⁺ to RNA, did they reproduce in vivo conditions?

Typographical errors:

Abstract line 21 "However, the discovery ... suggests the existence"

P 2 line 10 "synthesize m7G-RNA"

P4 line 36 "is a cofactor"

P6 line 29 "with the hypothesis"

P6 line 59 "in the active "

P7 line 19 "by an unknown"

P 7 line 41 "may be"

P7 line 52 "monomers or "

P 8 line 11 "the existence"

P8 line 24 "countered by a"

P8 line 17 10⁻³ is all superscript

P 10 line 18 "prokaryotes"

P 11 line 14 “thanks to the number”

P 11 line 40 “but the biological”

P 11 line 56 “by an unknown”

P 12 line 12 “by a variable”

P 12 line 31 “there exists dedicated”

References

Goodfellow, I. et al. (2005) ‘Calicivirus translation initiation requires an interaction between VPg and eIF4E’, *EMBO Reports*. European Molecular Biology Organization, 6(10), pp. 968–972. doi: 10.1038/sj.embor.7400510.

Wang, Y. et al. (2019) ‘NAD⁺-capped RNAs are widespread in the Arabidopsis transcriptome and can probably be translated’, *Proceedings of the National Academy of Sciences of the United States of America*. National Academy of Sciences, 116(24), pp. 12094–12102. doi: 10.1073/pnas.1903682116.

Zhang, H. et al. (2019) ‘NAD tagSeq reveals that NAD⁺-capped RNAs are mostly produced from a large number of protein-coding genes in Arabidopsis’, *Proceedings of the National Academy of Sciences of the United States of America*. National Academy of Sciences, 116(24), pp. 12072–12077. doi: 10.1073/pnas.1903683116.

Review form: Reviewer 2

Is the manuscript scientifically sound in its present form?

Yes

Are the interpretations and conclusions justified by the results?

Yes

Is the language acceptable?

Yes

Do you have any ethical concerns with this paper?

No

Have you any concerns about statistical analyses in this paper?

No

Recommendation?

Accept with minor revision (please list in comments)

Comments to the Author(s)

RNA can be subjected to diverse non-canonical 5' modifications – called capping. This review presents a comprehensive insight into the different types of non-canonical RNA modifications that exist. Non-canonical RNA capping can have reaching implications in gene expression and cellular physiological responses (in particular in response to stress) and this review provides an enzymology focused entry point into this exciting field of gene regulation. I recommend that the authors, where possible, provide details and/or relate their mechanistic descriptions of non-canonical capping of RNA to the physiological contexts they were originally described in. I have the following suggestion/queries:

- What role does, if any, non-canonical capping of RNA, has on small non-coding RNA biology? For example, are they stabilized by non-canonical capping or their affinity to RNA binding proteins altered?
- In the context of bacteria, do phages contribute to non-canonical RNA capping. If my memory serves me correctly, there are indirect evidence of this in T4 phage.
- Do non-canonical modification of RNA affect their spatiotemporal regulation?
- All the non-canonical capping modifications are given as enzymatic steps. This is fine; however, I would have preferred to have this as a schematic/cartoon to make this review article more accessible (e.g., like in Fig. 2).
- The text contains some typos and other grammatical errors that need to be rectified. For example: Page 3, line 10; Page 5, line 49/50.

Decision letter (RSOS-201979.R0)

Dear Dr Yuzenkova

On behalf of the Editors, we are pleased to inform you that your Manuscript RSOS-201979 "The expanding field of non- canonical RNA capping – new enzymes and mechanisms" has been accepted for publication in Royal Society Open Science subject to minor revision in accordance with the referees' reports. Please find the referees' comments along with any feedback from the Editors below my signature.

Please submit your revised manuscript and required files (see below) no later than 7 days from today's (ie 26-Mar-2021) date. Note: the ScholarOne system will 'lock' if submission of the revision is attempted 7 or more days after the deadline. If you do not think you will be able to meet this deadline please contact the editorial office immediately.

on behalf of Professor Xiaodong Zhang (Associate Editor) and Malcolm White (Subject Editor)
openscience@royalsociety.org

Associate Editor Comments to Author (Professor Xiaodong Zhang):

The manuscript has been reviewed by two experts who are supportive of the publication and have provided some valuable advice on ways to improve the accessibility and clarify of the manuscript. They have also identified some errors and inconsistencies that need to be addressed.

The authors are advised to take the recommendations fully to correct and improve the manuscript.

Reviewer comments to Author:

Reviewer: 1

Comments to the Author(s)

This review summarises current state of knowledge on the variety of non-canonical RNA caps discovered to date, the enzymes involved and their mechanisms, both in eukaryotes and in prokaryotes. The authors explain the mechanism of canonical RNA capping and then propose different non-canonical (NC) capping mechanisms. They address NC capping at the initiation of transcription and post-transcription and propose different enzymes that might be involved in these processes. Apart from the widely described NAD⁺/NADH, they also show some of the molecules recently identified as NC including dinucleotide polyphosphates, ADP ribose, and others. Finally, the authors state some of the questions regarding the mechanisms of NC capping and the physiological function of these alternative RNA caps that remain unanswered.

Main comments:

The article can be difficult to read for those who are not familiar with the enzymes and metabolites. Many enzymes are involved: a table can be useful to summarise which enzyme is supposed to be required for a particular non-canonical cap and what is the hypothetical mechanism of the non-canonical capping. There are many abbreviations used, and it might be helpful to also summarise these in the table.

Figures 1 and 3-8 are useful to understand the reactions but could be expanded beyond the simple text presented. Having more graphical figures, with 2D structures of the different interacting molecules and schemas for the protein involved, similar to the ones in Figure 2, might help the reader follow the different mechanisms explained in the Review.

Some positive sense ssRNA viruses have viral protein genome-linked (VPgs) covalently attached to the 5' end of their genome. In caliciviruses, VPgs were confirmed to interact with the cap-binding protein, eIF4E, and to be essential for translation (Goodfellow et al., 2005). Although they are not small molecules but proteins, it might be interesting to mention these viral alternatives as NC 5' RNA caps.

In the second paragraph of page 4 (or page 5 of 17 according to the enumeration at the top of each page). They mention that no NAD-capped RNAs were found in chloroplasts according to Wang et al. (2019), but Zhang et al. (2019) describe a NAD-capped chloroplast rRNA.

Then, they mention an experiment they performed with cyanobacterial RNAPs which could efficiently incorporate NAD⁺ into RNA (same page lines 16-20). It would be helpful if they explained a bit more their new findings (what proteins they used? what conditions? etc.) as they do with their experiments with NMNATs NadD and NadR in pages 5-6 (pages 6 of 17 and 7 of 17). (Wang et al., 2019) suggest the absence of NAD-capped RNA in chloroplasts could be due to

low levels of free NAD⁺ in this organelle. When they assessed if cyanobacterial RNAPs could incorporate NAD⁺ to RNA, did they reproduce in vivo conditions?

Typographical errors:

Abstract line 21 "However, the discovery ... suggests the existence"

P 2 line 10 "synthesize m7G-RNA"

P4 line 36 "is a cofactor"

P6 line 29 "with the hypothesis"

P6 line 59 "in the active "

P7 line 19 "by an unknown"

P 7 line 41 "may be"

P7 line 52 "monomers or "

P 8 line 11 "the existence"

P8 line 24 "countered by a"

P8 line 17 10⁻³ is all superscript

P 10 line 18 "prokaryotes"

P 11 line 14 "thanks to the number"

P 11 line 40 "but the biological"

P 11 line 56 "by an unknown"

P 12 line 12 "by a variable"

P 12 line 31 "there exists dedicated"

References

Goodfellow, I. et al. (2005) 'Calicivirus translation initiation requires an interaction between VPg and eIF4E', *EMBO Reports*. European Molecular Biology Organization, 6(10), pp. 968-972. doi: 10.1038/sj.embor.7400510.

Wang, Y. et al. (2019) 'NAD⁺-capped RNAs are widespread in the Arabidopsis transcriptome and can probably be translated', *Proceedings of the National Academy of Sciences of the United States of America*. National Academy of Sciences, 116(24), pp. 12094-12102. doi: 10.1073/pnas.1903682116.

Zhang, H. et al. (2019) 'NAD tagSeq reveals that NAD⁺-capped RNAs are mostly produced from a large number of protein-coding genes in Arabidopsis', *Proceedings of the National Academy of Sciences of the United States of America*. National Academy of Sciences, 116(24), pp. 12072-12077. doi: 10.1073/pnas.1903683116.

Reviewer: 2

Comments to the Author(s)

RNA can be subjected to diverse non-canonical 5' modifications – called capping. This review presents a comprehensive insight into the different types of non-canonical RNA modifications that exist. Non-canonical RNA capping can have reaching implications in gene expression and cellular physiological responses (in particular in response to stress) and this review provides an enzymology focused entry point into this exciting field of gene regulation. I recommend that the authors, where possible, provide details and/or relate their mechanistic descriptions of non-canonical capping of RNA to the physiological contexts they were originally described in. I have the following suggestion/queries:

- What role does, if any, non-canonical capping of RNA, has on small non-coding RNA biology? For example, are they stabilized by non-canonical capping or their affinity to RNA binding proteins altered?
- In the context of bacteria, do phages contribute to non-canonical RNA capping. If my memory serves me correctly, there are indirect evidence of this in T4 phage.
- Do non-canonical modification of RNA affect their spatiotemporal regulation?

- All the non-canonical capping modifications are given as enzymatic steps. This is fine; however, I would have preferred to have this as a schematic/cartoon to make this review article more accessible (e.g., like in Fig. 2).
- The text contains some typos and other grammatical errors that need to be rectified. For example: Page 3, line 10; Page 5, line 49/50.

===PREPARING YOUR MANUSCRIPT===

===PREPARING YOUR REVISION IN SCHOLARONE===

<https://royalsociety.org/journals/authors/author-guidelines/#supplementary-material> to include a suitable title and informative caption. An example of appropriate titling and captioning may be found at https://figshare.com/articles/Table_S2_from_Is_there_a_trade-off_between_peak_performance_and_performance_breadth_across_temperatures_for_aerobic_scops_in_teleost_fishes_/3843624.

Author's Response to Decision Letter for (RSOS-201979.R0)

See Appendix A.

Decision letter (RSOS-201979.R1)

Dear Dr yuzenkova,

It is a pleasure to accept your manuscript entitled "The expanding field of non- canonical RNA capping – new enzymes and mechanisms" in its current form for publication in Royal Society Open Science.

on behalf of Prof Malcolm White (Subject Editor)
openscience@royalsociety.org

Appendix A

We thank reviewers for their time and effort to read and provide comments on our admittedly not-easiest-to-read manuscript. We believe that incorporating their comments allowed us to improve our paper and make it more accessible for a general reader. Below are our point replies to reviewers' comments.

Reviewer: 1

Comments to the Author(s)

This review summarises current state of knowledge on the variety of non-canonical RNA caps discovered to date, the enzymes involved and their mechanisms, both in eukaryotes and in prokaryotes. The authors explain the mechanism of canonical RNA capping and then propose different non-canonical (NC) capping mechanisms. They address NC capping at the initiation of transcription and post-transcription and propose different enzymes that might be involved in these processes. Apart from the widely described NAD⁺/NADH, they also show some of the molecules recently identified as NC including dinucleotide polyphosphates, ADP ribose, and others. Finally, the authors state some of the questions regarding the mechanisms of NC capping and the physiological function of these alternative RNA caps that remain unanswered.

Main comments:

The article can be difficult to read for those who are not familiar with the enzymes and metabolites. Many enzymes are involved: a table can be useful to summarise which enzyme is supposed to be required for a particular non-canonical cap and what is the hypothetical mechanism of the non-canonical capping.

We summarised the known and potential capping enzymes and products of their enzymatic functions in the Table 2. Mechanism of action is described in the main text and is not included in this table due to being too extensive.

There are many abbreviations used, and it might be helpful to also summarise these in the table.

We summarised abbreviations in a new Table 1.

Figures 1 and 3-8 are useful to understand the reactions but could be expanded beyond the simple text presented. Having more graphical figures, with 2D structures of the different interacting molecules and schemas for the protein involved, similar to the ones in Figure 2, might help the reader follow the different mechanisms explained in the Review.

We included new 2D cartoon figures of all presented mechanisms in the text.

Some positive sense ssRNA viruses have viral protein genome-linked (VPgs) covalently attached to the 5' end of their genome. In caliciviruses, VPgs were confirmed to interact with the cap-binding protein, eIF4E, and to be essential for translation (Goodfellow et al.,

2005). Although they are not small molecules but proteins, it might be interesting to mention these viral alternatives as NC 5' RNA caps.

We included new paragraph about VPgs (page 22)

In the second paragraph of page 4 (or page 5 of 17 according to the enumeration at the top of each page). They mention that no NAD-capped RNAs were found in chloroplasts according to Wang et al. (2019), but Zhang et al. (2019) describe a NAD-capped chloroplast rRNA.

We corrected the information about chloroplast RNAP capping in the text (page 6)

Then, they mention an experiment they performed with cyanobacterial RNAPs which could efficiently incorporate NAD⁺ into RNA (same page lines 16-20). It would be helpful if they explained a bit more their new findings (what proteins they used? what conditions? etc.) as they do with their experiments with NMNATs NadD and NadR in pages 5-6 (pages 6 of 17 and 7 of 17). (Wang et al., 2019) suggest the absence of NAD-capped RNA in chloroplasts could be due to low levels of free NAD⁺ in this organelle. When they assessed if cyanobacterial RNAPs could incorporate NAD⁺ to RNA, did they reproduce in vivo conditions?

We include primary data on transcription by various RNAPs including cyanobacterial one. Reported levels of NAD⁺/NADH vary depending on conditions, but maximum concentration in chloroplasts are not massively different from maximum concentration in cytosol. According to recent review (Gakiere et al., 2018), total for (NAD⁺+NADH) are 0.39 mM for chloroplast and 0.65 mM for cytosol. These numbers are in the range of Km for NAD⁺/NADH incorporation in a transcript (around 0.4 mM), admittedly by E. coli enzyme.

We performed our reactions using 0.5 mM NAD⁺, thus we reproduced most favourable in vivo conditions. The main aim of the experiment was to show that initiation of transcription with cofactors is a ubiquitous feature among prokaryotic enzymes, and that cyanobacterial RNAP (and quite possibly chloroplast one by proxy) does not have any structural domains which precludes incorporation of cofactors.

We clarified other reaction conditions we used for cyanobacterial transcription.

Typographical errors:

Abstract lien 21 "However, the discovery ... suggests the existence"

P 2 line 10 "synthesize m7G-RNA"

P4 line 36 "is a cofactor"

P6 line 29 "with the hypothesis"

P6 line 59 "in the active "

P7 line 19 "by an unknown"

P 7 line 41 "may be"

P7 line 52 "monomers or "

P 8 line 11 "the existence"
P8 line 24 "countered by a"
P8 line 17 10⁻³ is all superscript
P 10 line 18 "prokaryotes"
P 11 line 14 "thanks to the number"
P 11 line 40 "but the biological"
P 11 line 56 "by an unknown"
P 12 line 12 "by a variable"
P 12 line 31 "there exists dedicated"

Typographical errors were corrected

References

- Goodfellow, I. et al. (2005) 'Calicivirus translation initiation requires an interaction between VPg and eIF4E', EMBO Reports. European Molecular Biology Organization, 6(10), pp. 968–972. doi: 10.1038/sj.embor.7400510.
- Wang, Y. et al. (2019) 'NAD⁺-capped RNAs are widespread in the Arabidopsis transcriptome and can probably be translated', Proceedings of the National Academy of Sciences of the United States of America. National Academy of Sciences, 116(24), pp. 12094–12102. doi: 10.1073/pnas.1903682116.
- Zhang, H. et al. (2019) 'NAD tagSeq reveals that NAD⁺-capped RNAs are mostly produced from a large number of protein-coding genes in Arabidopsis', Proceedings of the National Academy of Sciences of the United States of America. National Academy of Sciences, 116(24), pp. 12072–12077. doi: 10.1073/pnas.1903683116.

Reviewer: 2

Comments to the Author(s)

RNA can be subjected to diverse non-canonical 5' modifications – called capping. This review presents a comprehensive insight into the different types of non-canonical RNA modifications that exist. Non-canonical RNA capping can have reaching implications in gene expression and cellular physiological responses (in particular in response to stress) and this review provides an enzymology focused entry point into this exciting field of gene regulation. I recommend that the authors, where possible, provide details and/or relate their mechanistic descriptions of non-canonical capping of RNA to the physiological contexts they were originally described in. I have the following suggestion/queries:

- What role does, if any, non-canonical capping of RNA, has on small non-coding RNA biology? For example, are they stabilized by non-canonical capping or their affinity to RNA binding proteins altered?

The data on stability are conflicting – in E. coli it was shown that capping increased stability of one specific RNA; in eukaryotes it was researched more extensively, and current view is that capping targets RNA for degradation.

We summarise briefly the emerging physiological role of NC caps. But this topic is beyond the scope of this review and was recently reviewed e.g. in

Vasilyev et al. (2019) Non-canonical features and modifications on the 5'- end of bacterial sRNAs and mRNAs Wiley Interdiscip Rev RNA 10 (2) e1509

Julius, C., and Yuzenkova, Y. (2019) Noncanonical RNA-capping: Discovery, mechanism, and physiological role debate. Wiley Interdiscip Rev RNA 10 (2) e1512.

- In the context of bacteria, do phages contribute to non-canonical RNA capping. If my memory serves me correctly, there are indirect evidence of this in T4 phage.

We are not aware of any published data on this; T4 does not have its own RNAP, and relies on E. coli host RNAP to transcribe its genes. Since E. coli RNAP is able to cap transcripts with cofactors, it follows that there is high probability of T4 transcripts being capped.

NAD⁺ plays role in a developmental programme of T4 phage in a distinct way – T4 codes for three mono-ADP-ribosyltransferases which transfer ADP-ribose from NAD⁺ to a target. One of them, Alt is present in a phage particle. These enzymes modify a number of E. coli proteins, including RNAP, EF-TU, toxin-antitoxin, etc. Potentially these enzymes can modify nucleic acids.

Single-subunit RNAP of T7 phage efficiently incorporates NAD⁺ into the transcript in vitro on +1A promoters, yet T7 strongly prefers +1G promoters in vivo (there are class II phage promoters where to some extent initiation can start with ATP or NAD). Altogether, there is a probability to find capped T7 RNA species, but they will be low abundant.

- Do non-canonical modification of RNA affect their spatiotemporal regulation?

This is very exciting proposition, which we would like to test, as so many proteins bind NAD as a cofactor and therefore would drag capped RNA to their specific subcellular location, but currently not much is known about it.

- All the non-canonical capping modifications are given as enzymatic steps. This is fine; however, I would have preferred to have this as a schematic/cartoon to make this review article more accessible (e.g., like in Fig. 2).

We included new 2D cartoon figures of all presented mechanisms in the text (Fig. 1 and 3-9).

- The text contains some typos and other grammatical errors that need to be rectified. For example: Page 3, line 10; Page 5, line 49/50.

Typographical errors were corrected.